# Empagliflozin and Liraglutide Differentially Modulate Cardiac Metabolism in Diabetic Cardiomyopathy in Rats

**DOI:** 10.3390/ijms22031177

**Published:** 2021-01-25

**Authors:** Nguyen Ngoc Trang, Cheng-Chih Chung, Ting-Wei Lee, Wan-Li Cheng, Yu-Hsun Kao, Shih-Yu Huang, Ting-I Lee, Yi-Jen Chen

**Affiliations:** 1International Ph.D. Program in Medicine, College of Medicine, Taipei Medical University, Taipei 11031, Taiwan; drnguyenngoctrang@gmail.com; 2Cardiovascular Research Center, Wan Fang Hospital, Taipei Medical University, Taipei 11696, Taiwan; michaelchung110@gmail.com (C.-C.C.); wanlicheng80@gmail.com (W.-L.C.); yuhsunkao@gmail.com (Y.-H.K.); yjchen@tmu.edu.tw (Y.-J.C.); 3Division of Cardiology, Department of Internal Medicine, School of Medicine, College of Medicine, Taipei Medical University, Taipei 11031, Taiwan; 4Division of Endocrinology and Metabolism, Department of Internal Medicine, School of Medicine, College of Medicine, Taipei Medical University, Taipei 11031, Taiwan; b8801138@tmu.edu.tw; 5Division of Endocrinology and Metabolism, Department of Internal Medicine, Wan Fang Hospital, Taipei Medical University, Taipei 11696, Taiwan; 6Graduate Institute of Clinical Medicine, College of Medicine, Taipei Medical University, Taipei 11031, Taiwan; 7Division of Cardiology, Department of Internal Medicine, Cathay General Hospital, Taipei 10630, Taiwan; sam1104h@gmail.com; 8Department of General Medicine, School of Medicine, College of Medicine, Taipei Medical University, Taipei 11031, Taiwan

**Keywords:** diabetic cardiomyopathy, glucagon-like peptide 1 receptor agonists, liraglutide, sodium-glucose cotransporter-2 inhibitors, empagliflozin, fatty acid and glucose metabolism, inflammation

## Abstract

Glucagon-like peptide 1 receptor agonists (GLP-1RAs) and sodium-glucose cotransporter-2 inhibitors (SGLT2is) are antihyperglycemic agents with cardioprotective properties against diabetic cardiomyopathy (DCM). However, the distinctive mechanisms underlying GLP-1RAs and SGLT2is in DCM are not fully elucidated. The purpose of this study was to investigate the impacts of GLP1RAs and/or SGLT2is on myocardial energy metabolism, cardiac function, and apoptosis signaling in DCM. Biochemistry and echocardiograms were studied before and after treatment with empagliflozin (10 mg/kg/day, oral gavage), and/or liraglutide (200 μg/kg every 12 h, subcutaneously) for 4 weeks in male Wistar rats with streptozotocin (65 mg/kg intraperitoneally)-induced diabetes. Cardiac fibrosis, apoptosis, and protein expression of metabolic and inflammatory signaling molecules were evaluated by histopathology and Western blotting in ventricular cardiomyocytes of different groups. Empagliflozin and liraglutide normalized myocardial dysfunction in diabetic rats. Upregulation of phosphorylated-acetyl coenzyme A carboxylase, carnitine palmitoyltransferase 1β, cluster of differentiation 36, and peroxisome proliferator-activated receptor-gamma coactivator, and downregulation of glucose transporter 4, the ratio of phosphorylated adenosine monophosphate-activated protein kinase α2 to adenosine monophosphate-activated protein kinase α2, and the ratio of phosphorylated protein kinase B to protein kinase B in diabetic cardiomyocytes were restored by treatment with empagliflozin or liraglutide. Nucleotide-binding oligomerization domain, leucine-rich repeat and pyrin domain-containing 3, interleukin-1β, tumor necrosis factor-α, and cleaved caspase-1 were significantly downregulated in empagliflozin-treated and liraglutide-treated diabetic rats. Both empagliflozin-treated and liraglutide-treated diabetic rats exhibited attenuated myocardial fibrosis and apoptosis. Empagliflozin modulated fatty acid and glucose metabolism, while liraglutide regulated inflammation and apoptosis in DCM. The better effects of combined treatment with GLP-1RAs and SGLT2is may lead to a potential strategy targeting DCM.

## 1. Introduction

The prevalence of diabetes mellitus (DM) is increasing worldwide with estimates increasing from 463 million people in 2019 to 700 million people by 2045 [1]. Cardiovascular (CV) complications are considered to be the major causes of mortality in DM patients [2]. Compared with non-DM patients, a higher incidence of myocardial dysfunction is found in DM patients [3]. DM doubles the risk of CV diseases [4], and about 75% of deaths in DM are due to coronary artery disease [5]. Although advances in medical management and lifestyle interventions have reduced CV mortality in DM patients by about 40% over the last decade, the actual number of deaths is predicted to rise as a result of increases in DM incidence and in aging populations [6].

The year 2015 represented a marked transformation in the treatment paradigm for Type 2 DM with CV diseases as demonstrated from the results of the sodiumglucose cotransporter inhibitor (SGLT2is) empagliflozin [7], and the glucagon-like peptide (GLP)-1 receptor agonists (GLP-1RAs) liraglutide and semaglutide, [8] all of which illustrated CV benefits. SGLT2is and GLP-1RAs have emerged as new classes of antihyperglycemic agents that also reduce CV risks and promote significant improvements in cardiac contractile function [9]. Several mechanisms were postulated for the CV effects of SGLT2is, including effects on osmotic diuresis and natriuresis contributing to lowering of blood pressure (BP), decreases in arterial stiffness and vascular resistance, improvements in weight and visceral adiposity, and a shift in myocardial fuel energetics [10]. Meanwhile, potential mechanisms mediating the beneficial effects of GLP-1RAs in reducing CV events include reductions in body weight (BW), BP, and lipoproteins and interactions with GLP-1 receptors in the CV system, resulting in improvements of endothelial function and cardiac function during coronary ischemia, as well as anti-inflammatory and anti-atherosclerotic effects [11].

Although SGLT2is and GLP-1RAs have significant cardiac impacts in clinical practice, the mechanisms by which SGLT2is and GLP-1RAs improve CV outcomes are highly speculative and not fully understood. SGLT2is and GLP-1RAs have not been well compared simultaneously in a DM model. In addition, it is not clear whether combinations of GLP-1RAs and SGLT2is could lead to distinctive impacts on DM cardiomyopathy (DCM). The purposes of this study were to investigate the impacts of a GLP-1RA and an SGLT2i on myocardial energy metabolism, cardiac function, and apoptosis signaling in DCM and evaluate the combined myocardial effects of a GLP-1RA and an SGLT2i in DM rats.

## 2. Results

### 2.1. Effects of Empagliflozin and Liraglutide on Biochemistry, BP, Myocardial Function, and Heart Size

Compared with control rats, DM rats, empagliflozin-treated DM rats, and liraglutide-treated DM rats all had higher blood glucose levels. However, empagliflozin-treated DM rats and liraglutide-treated DM rats had lower levels of blood glucose than did DM rats (Table 1). Besides, empagliflozin-treated DM rats had lower levels of blood glucose than did liraglutide-treated DM rats.

At 16 weeks of age, triglycerides (TGs) were attenuated by treatment with empagliflozin compared with DM rats. Total cholesterol (TC) and high-density lipoprotein cholesterol (HDL-C) were similar among the four groups. The low-density lipoprotein cholesterol (LDL-C) of the DM group was higher compared with the control group. Both empagliflozin-treated DM rats and liraglutide-treated DM rats had attenuated levels of LDL-C. Additionally, free fatty acids (FFAs) were significantly elevated in the DM group compared with the control group. FFAs were significantly reversed in empagliflozin-treated DM rats and liraglutide-treated DM rats. Both the empagliflozin-treated DM and liraglutide-treated DM groups additionally had normalized concentrations of suppression of tumorgenicity-2 (ST2) compared with the DM group.

Systolic and diastolic BPs after treatment were similar among the four groups. However, DM rats had a lower heart rate than the control. However, there were no significant differences among control rats, empagliflozin-treated DM rats, and liraglutide-treated DM rats at 16 weeks of age.

BWs were similar among the four groups before treatment. However, lower BWs were noted in the DM, empagliflozin-treated DM, and liraglutide-treated DM groups compared with the control group. BWs were similar between DM rats and liraglutide-treated DM rats, but empagliflozin-treated DM rats had higher BWs after treatment than did DM rats. In addition, DM rats had a greater heart-to-body weight ratio than control, empagliflozin-treated DM, and liraglutide-treated DM rats.

As shown in Figure 1, at 16 weeks of age, DM rats had a higher left ventricular end-diastolic diameter (LVEDd), left ventricular end-systolic diameter (LVESd), end-diastolic volume (EDV), and end-systolic volume (ESV), and a lower ejection fraction (EF) and fractional shortening (FS) value compared with control rats, empagliflozin-treated DM rats, and liraglutide-treated DM rats. These values were similar between empagliflozin-treated DM rats and liraglutide-treated DM rats.

### 2.2. Effects of Empagliflozin and Liraglutide on Myocardial Fatty Acid Metabolism

Both empagliflozin-treated DM rats and liraglutide-treated DM rats had higher ratio protein expressions of phosphorylated 5’ adenosine monophosphate-activated protein kinase α2 (pAMPKα2) to 5’adenosine monophosphate-activated protein kinase α2 (AMPKα2) compared with DM rats (Figure 2b). In addition, phosphorylated-acetyl coenzyme A carboxylase (pACC) protein expression in DM rats was higher than those of control, liraglutide-treated DM, and empagliflozin-treated DM rats (Figure 2c). Moreover, empagliflozin-treated DM rats had lower pACC protein level compared with liraglutide-treated DM rats. Peroxisome proliferator-activated receptor-gamma coactivator (PGC)-1α protein expression was lower in empagliflozin-treated DM rats and liraglutide-treated DM rats compared with DM rats (Figure 2d). Carnitine palmitoyltransferase (CPT)-1β and the cluster of differentiation (CD)-36 protein levels were lower in the empagliflozin-treated DM group and liraglutide-treated DM group compared with the DM group (Figure 2e,f).

### 2.3. Effects of Empagliflozin and Liraglutide on Myocardial Glucose Metabolism

The control group, liraglutide-treated DM group, and empagliflozin-treated DM group had higher glucose transporter 4 (GLUT4) protein expression than did the DM group (Figure 3b). In addition, the ratio of phosphorylated insulin receptor substrate 1 (pIRS1) (Ser 307) to insulin receptor substrate 1 (IRS1) was significantly higher in empagliflozin-treated DM rats and liraglutide-treated DM rats than in DM rats (Figure 3c). Similarly, both empagliflozin-treated DM rats and liraglutide-treated DM rats had an increased the ratio of phosphorylated protein kinase B (pAkt) (Ser 473) to protein kinase B (Akt) compared with DM rats (Figure 3d).

### 2.4. Effects of Empagliflozin and Liraglutide on Myocardial Inflammatory Cytokines

Figure 4a represents the NLR NOD-like receptor family pyrin domain-containing 3 (NLRP3) inflammasome pathway. NLRP3 protein levels were significantly decreased in empagliflozin-treated DM rats and liraglutide-treated DM rats compared with DM rats. Protein expressions of pro-caspase-1 and cleaved caspase-1 were normalized by empagliflozin and liraglutide treatment. Besides, interleukin (IL)-1β protein expression was lower in the empagliflozin-treated DM group and liraglutide-treated DM group than in the DM group. Additionally, protein expression of tumor necrosis factor (TNF)-α (Figure 4b) was lower in the empagliflozin-treated DM group and liraglutide-treated DM group than in the DM group.

### 2.5. Effects of Empagliflozin and Liraglutide on Myocardial Fibrosis and Apoptosis

The ratio of phosphorylated signal transducer and activator of transcription 3 (pSTAT3) to signal transducer and activator of transcription 3 (STAT3) (Figure 5b) were significantly higher in empagliflozin-treated and liraglutide-treated DM groups than in the DM group. Moreover, liraglutide-treated DM rats had a higher pSTAT3/STAT3 compared with empagliflozin-treated DM rats. Both empagliflozin-treated and liraglutide-treated DM hearts exhibited significantly decreased protein levels of phosphorylated extracellular signal-regulated kinase 1/2 (pERK1/2) (Figure 5c) compared with DM rats. In addition, the apoptotic index with immunohistochemical staining of caspase-3 in DM ventricular cardiomyocytes was higher than that of control cardiomyocytes. Both empagliflozin-treated and liraglutide-treated DM cardiomyocytes had a decreased apoptotic index (Figure 5d). DM hearts had 2.5-fold higher fibrosis than control hearts (Figure 5e,f). Fibrosis and apoptosis in DM cardiomyocytes were more attenuated with liraglutide treatment than with empagliflozin treatment.

### 2.6. Combined Effects of Empagliflozin and Liraglutide in DCM

We further evaluated the combined effects of GLP-1RA and SGLT2i in DM rats. As shown in Table 2, combination-treated DM and empagliflozin-treated DM groups had similar echocardiographic values, which were better than those of liraglutide-treated DM rats. There was no difference in the heart rates, BP, HWs, heart-to-body weight ratios, and the biochemical values of the combination-treated DM group versus the groups treated with the individual compounds. The combination-treated DM rats had higher BWs and lower fasting blood glucose (FBS) than liraglutide-treated DM rats. Moreover, the fibrosis area of combination-treated DM rats was similar to that in liraglutide-treated DM rats but lesser than that of empagliflozin-treated DM rats.

## 3. Discussion

DCM decreased myocardial function, characterized by structural and functional abnormalities, including left ventricle (LV) hypertrophy, fibrosis, and cell signaling abnormalities [12]. Our study compared the effects of empagliflozin and liraglutide, which belong to two new classes of anti-hyperglycemic agents, as cardio-protection in DCM, and we found that the FBS value of the liraglutide group was statistically different from the FBS value of the empagliflozin group, so empagliflozin seems to be more efficient at reducing glycemia. Hyperglycemia impairs systolic and diastolic functions [13]. Similar to our previous studies [14,15], we also found a significantly dilated LV chamber with a lower EF and shorter FS in DM hearts. However, DM rats treated with empagliflozin or liraglutide showed significant improvements in myocardial systolic function. Like previous findings [14,16], empagliflozin or liraglutide produced significant improvements in cardiac functions, suggesting that these anti-hyperglycemic agents might be beneficial in treating DCM.

5’ adenosine monophosphate-activated protein kinase (AMPK) plays an important role in regulating myocardial energy metabolism; ameliorating vascular endothelial dysfunction, inflammation, and apoptosis; and regulating autophagy [17]. Interestingly, Balteau et al. also found that GLP1 concomitantly induced AMPK activation and prevented high glucose concentration-mediated reactive oxygen species (ROS) production [18]. The decrease in the myocardial pAMPKα2 protein level in our DM group might be associated with an excessive oversupply of lipids that inactivate AMPKα2 [19]. In our investigation, we found that empagliflozin- or liraglutide-treated DM rats exhibited significant stimulation of myocardial pAMPKα2 protein expression. The activated form of AMPK was expected to induce phosphorylation of acetyl coenzyme A carboxylase (ACC). However, our results showed the opposite modulation. The previous study also showed an activated AMPK with suppressed ACC activity in metformin-treated hepatocytes [20]. Additional studies will be required to further elucidate the precise mechanism underlying our findings. In addition, AMPK activates PGC-1α expression and directly enhances its activity through phosphorylation of AMPK [21]. PGC-1α plays a vital part in modulating fatty acid (FA) β-oxidation and myocardial lipid over-accumulation that might worsen cardiac lipid and glucose utilization and the energy balance [22]. Previous studies illustrated that normalizing FA metabolism in DM hearts reversed impaired myocardial contractility [23,24]. Interestingly, we observed that myocardial FA regulator substrates of pACC, CD36, and CPT-1β were augmented after treatment with empagliflozin or liraglutide. The increase in the FA transporter protein CD36 in DM hearts may expedite the uptake of intracellular FAs, as evidenced by elevated pACC protein levels. AMPK is also involved in the delivery of FAs to cardiomyocytes through its regulation of the FA transporter CD36 [25]. The effects of empagliflozin and liraglutide on myocardial lipid metabolism suggest a cardioprotective potential of these two anti-hyperglycemic agents through switching from FFAs.

This study found that DM hearts treated with liraglutide or empagliflozin restored the ratios of pIRS1 (Ser 307)/IRS1 and pAkt (Ser 473)/Akt. However, it is not clear whether these effects may play a role in cardiac function and metabolic homeostasis in these animals. Our results showed that the GLUT4 protein expression significantly increased with empagliflozin or liraglutide treatment compared with the DM rat hearts. However, an increased protein level is not enough to hypothesize increased GLUT4 expression on the cell membrane, since membrane translocation was not analyzed in this study.

Increased inflammation and oxidative stress due to DM were found to precipitate atherosclerosis and activate cytokine secretions [26], which may result in the development of cardiomyopathy. In this study, liraglutide or empagliflozin mitigated myocardial IL-1β protein levels in DM rats. We also found that NLRP3 expression significantly decreased in both empagliflozin-treated DM rats and liraglutide-treated DM rats compared to DM rats. In addition, caspase-1 (an inflammasome component) was also decreased in empagliflozin-treated DM rats and liraglutide-treated DM rats. Zhu et al. found that the expression of NLRP3 inflammasome components (including NLRP3 and caspase-1) also substantially decreased in the livers of mice after treatment with liraglutide [27]. 

GLP-1RA treatment protects cardiomyocytes from apoptosis, and the protective effects were attenuated by phosphoinositide 3-kinase (PI3K) and/or extracellular signal-regulated kinase 1/2 (ERK1/2) inhibition [28]. Our investigation also found a significant reduction in pERK1/2 protein expression in empagliflozin- or liraglutide-treated DM rats. Activation of STAT3 has been suggested to reduce cardiac fibrosis or hypertrophy in DM via the inhibition of apoptosis or an increase of antioxidants [29], although STAT3 may promote proliferation and collagen production in isolated cardiac fibroblasts treated with a high concentration of glucose [30]. In this study, consistent with the previous studies [31,32], DM rats had decreased p-STAT3 compared with controls. Moreover, an increase in the ratio of pSTAT3/STAT3 protein expression in empagliflozin-treated DM rats and liraglutide-treated DM rats was seen. These findings suggest the potential role of STAT3 regulation in cardiac remodeling in empagliflozin- or liraglutide-treated DM rats. In addition, our apoptotic index with cleaved caspase-3 decreased with empagliflozin or liraglutide treatment, which demonstrated a cardioprotective role. Increased cardiac apoptosis is a major risk factor for the development of DCM [33]. Our experimental data also illustrated the promising CV therapeutic benefits of empagliflozin and liraglutide by switching FA oxidation and glucose metabolism, and their anti-inflammatory, antiapoptotic, and antifibrotic effects. Figure 6 demonstrates the potential effects of empagliflozin (SGLT2i) and liraglutide (GLP-1RA) in DM hearts in this study. Empagliflozin modulates FA and glucose metabolism, while liraglutide regulates inflammation, fibrosis, and apoptosis in DCM. 

This study has some limitations. First, our results showed that the combined treatment with GLP-1RA and SGLT2i may lead to a potential novel strategy in targeting DCM, but the effects of empagliflozin and liraglutide on the myocardial ATP content, cytokine analysis, NLRP3 signaling, cardiac metabolism, and collagen expression in DCM remain unclear. Besides, the mechanism underlying the superiority of the combined treatment needs further study. Second, our findings may not fully translate to DM patients in clinics, since the treatment already started 2 weeks after streptozotocin (STZ) injection. Third, the animal numbers used for this study were relatively low and the statistical power may be underestimated, even though Duncan’s method was chosen for post hoc analysis. Finally, we investigated the local inflammatory responses by using Western blot analysis but not ELISA to measure the cardiac expression of inflammatory cytokine in DCM. Thus, our results may be relatively semi-quantitative.

In this study, for the first time, we found that treatment of DM rats with the combination of GLP-1RA and SGLT2i may produce better FS than treatment with GLP-1RA alone. Although the mechanisms underlying our findings were not clear, we speculated that the combination of the anti-fibrosis potential of GLP-1RA and the hemodynamic modulation or decreases in pACC from SGLT2i may contribute to better cardiac function.

## 4. Methods

### 4.1. Animal Models and Tissue Preparations

This investigation conformed to the institutional *Guide for the Care and Use of Laboratory Animals* and the *Guide for the Care and Use of Laboratory Animals* published by the US National Institutes of Health, and was approved by the Institutional Animal Care and Use Committee of Taipei Medical University (LAC-2019-0364). To induce DM, some of the 10-week-old male Wistar rats (~335 ± 4.5 g) received a single intraperitoneal injection of streptozotocin (STZ) (65 mg/kg, Sigma-Aldrich, St. Louis, MO, USA) after 10 h of overnight starvation [34,35]. DM was diagnosed according to a high fasting plasma glucose level (≥15 mmol/L) as measured with a glucometer (Bayer Breeze 2 glucometer, Bayer Health Care, Mishawaka, IN, USA). Two weeks after DM induction (at 12 weeks of age), DM rats were randomly assigned to receive empagliflozin (10 mg/kg/day, oral gavage; Jardiance, Boehringer Ingelheim Pharmaceuticals, Ridgefield, CT, USA) [36], liraglutide (200 μg/kg body weight every 12 h subcutaneously; Novo Nordisk, Bagsvaerd, Denmark) [37], combined treatment with empagliflozin and liraglutide; or the vehicle (1 mL normal saline once daily by oral gavage) for 4 weeks. Rats were anesthetized by deep inhalation with 5% isoflurane [38] and sacrificed at 16 weeks of age. BWs were measured prior to euthanasia. Each heart was rapidly excised, weighed, and dissected. Cardiac tissues were rinsed in a cold physiological saline solution. Freshly isolated ventricular tissues were rinsed in a cold physiological saline solution and frozen in liquid nitrogen for protein isolation. Five ventricular preparations from each group were placed in formalin for histological examinations.

### 4.2. Blood Sampling and Biochemical Measurements

Blood samples were collected from rats at 10 weeks of age before treatment and at 16 weeks of age prior to sacrifice into collecting vials, and serum was prepared and stored at −20 °C pending further analysis. Serum samples were used to measure the following parameters by means of specific kits: fasting serum glucose, TGs, TC, HDL-C, LDL-C, FFAs, and ST2.

### 4.3. Hemodynamic Studies: BP and Echocardiogram Measurements

The systolic BP, diastolic BP, and heart rate were measured with a non-invasive blood pressure monitor tail-cuff method for rats (MK-2000, Muromachi Kikai, Tokyo, Japan) [39]. The BP and heart rate were monitored at 10 weeks of age and at 16 weeks prior to sacrifice. An echocardiogram was performed using the Vivid I ultrasound cardiovascular system (GE Healthcare, Haifa, Israel) at 10 weeks of age and at 16 weeks prior to sacrifice. The following cardiac structures were measured using M-Mode tracing of the LV: LVEDd, LVESd, EDV, ESV, EF, and FS values, as described previously [40].

### 4.4. Histopathological Examination of Cardiomyocytes

Cardiac fibrosis was assessed on slides stained with Masson’s trichrome according to previously described protocols [41]. Collagen was stained blue, and myocytes were stained red. Whole hearts were scanned at 20× magnification on a high-resolution microscope using Case Viewer software (3DHistech Kft., Budapest, Hungary). The fibrosis content was quantified by identifying and counting the number of blue-stained pixels as a percentage of the total LV tissue area using ImageJ1.53e (National Institutes of Health, Bethesda, MD, USA).

*Caspase-3* expression was determined according to a conventional immunohistochemical method [42]. Briefly, heart sections (5 μm) were allowed to equilibrate to room temperature and exposed to acetone for 10 min before initiating the streptavidin–biotin staining technique (Vectastain Universal Quick Kit; Vector Laboratories, CA, USA). Sections were then preblocked with 10% horse serum/phosphate-buffered saline (PBS) + 0.2% Tween 20 Tris-buffered saline (TBS) for 10 min at room temperature, followed by elimination of excess serum from the section and incubation with a specific antibody and an isotype-matched control antibody at appropriate dilutions. The primary antibody cleaved-*caspase-3* (1:400 dilution; Cell Signaling, MA, USA) was incubated with the sections. After washing, bound antibodies were detected by a universal biotinylated antibody prediluted in TBS at room temperature for 20 min, followed by incubation for 10 min with peroxidase-conjugated streptavidin. Sections were rinsed, counterstained, and mounted in an aqueous mounting medium. Sections were scanned at 20× magnification on a high-resolution microscope using Case Viewer software (3DHistech Kft., Budapest, Hungary). The degree of apoptosis was determined by the apoptotic index (%) = (number of (+) stained myocyte nuclei/total number of myocyte nuclei counted) × 100.

### 4.5. Western Blot Analysis 

As described previously [34], equal amounts of proteins were resolved by sodium dodecyl sulfate-polyacrylamide gel electrophoresis (SDS-PAGE), followed by electrophoretic transfer of proteins onto nitrocellulose membranes. Blots were probed with antibodies against AMPKα2 (Cell Signaling Technology), pAMPKα2 (Millipore, St. Louis, MO, USA), pACC (Millipore, St. Louis, MO, USA), PGC-1α (Abcam, Cambridge, MA, USA), CPT-1β (Abcam, Cambridge, MA, USA), CD36 (Santa Cruz Biotechnology, CA, USA), GLUT4 (Abcam, Cambridge, MA, USA), pIRS1 (Ser 307, Cell Signaling Technology), pAkt (Ser 473, Cell Signaling Technology), PI3K/Akt (Cell Signaling Technology), NLRP3 (Novus Biologicals), TNF-α (Santa Cruz Biotechnology, CA, USA), IL-1β (Cell Signaling Technology), pro-caspase-1 (Novus Biologicals), cleaved caspase-1 (Novus Biologicals), STAT3 (Cell Signaling Technology), pSTAT3 (Cell Signaling Technology), pERK1/2 (Cell Signaling Technology), and secondary antibodies conjugated with horseradish peroxidase (Leinco Technology, St. Louis, MO, USA). Bound antibodies were detected using an enhanced chemiluminescence detection system (Millipore, St. Louis, MO, USA) and analyzed with AlphaEaseFC software (Alpha Innotech, San Leandro, CA, USA). Targeted bands were normalized to cardiac glyceraldehyde 3-phosphate dehydrogenase (GAPDH, Sigma-Aldrich) to confirm equal protein loading.

### 4.6. Statistical Analysis

All quantitative data are expressed as the mean ± standard error of the mean (SEM). Statistical significance between different groups was determined using a one-way analysis of variance (ANOVA) with Duncan’s method in SigmaPlot version 14 (Systat Software, San Jose, CA, USA) for multiple comparisons as appropriate. A value of *p* < 0.05 was considered statistically significant.

## 5. Conclusions

Both empagliflozin and liraglutide have cardioprotective effects by increasing the systolic function of DM rats. Empagliflozin and liraglutide improved heart function in DM rats by switching fatty acid oxidation and glucose metabolism, and through anti-inflammatory, anti-apoptosis, and fibrosis effects. The better effects of combined treatment with GLP-1RA and SGLT2i may potentially lead to a novel strategy for targeting DM cardiomyopathy.

## Figures and Tables

**Figure 1 ijms-22-01177-f001:**
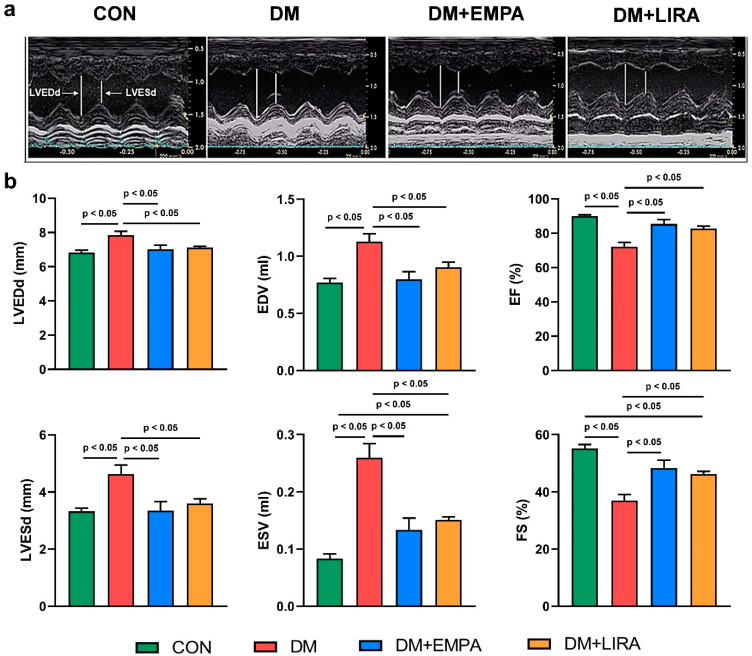
Echocardiograms of control (CON), diabetes mellitus (DM), empagliflozin-treated DM (DM + EMPA) rats, and liraglutide-treated DM (DM + LIRA) rats at 16 weeks of age. (**a**) Representative M-mode echocardiograms of CON (*N* = 8), DM (*N* = 8), DM + EMPA (*N* = 8), and DM + LIRA rats (*N* = 8). (**b**) Bar graphs of echocardiographic measurement results of left ventricle (LV) end-diastolic and end-systolic diameters (LVEDd and LVEDs), end-diastolic volume (EDV), end-systolic volume (EDV), ejection fraction (EF), and fraction shortening (FS).

**Figure 2 ijms-22-01177-f002:**
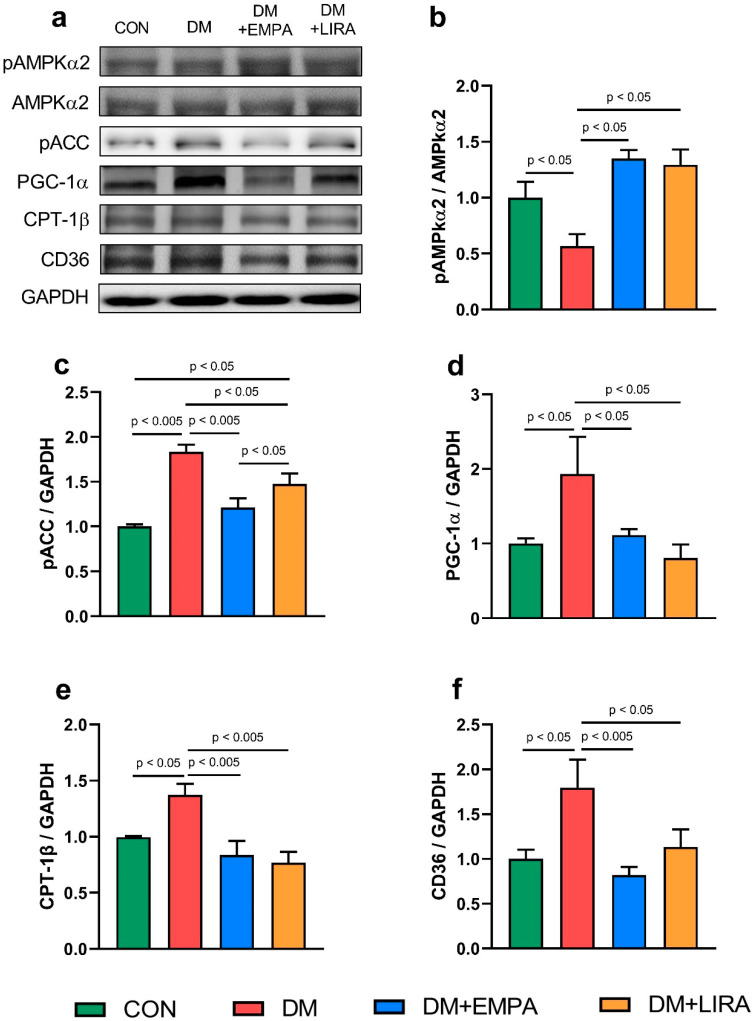
Expressions of cardiac fatty acid regulatory proteins in control (CON) and diabetic (DM) rats with and without treatment with empagliflozin (EMPA) and liraglutide (LIRA). (**a**) Representative immunoblot image. (**b**) Ratio of phosphorylated adenosine monophosphate-activated protein kinase α2 (pAMPKα2) to 5’adenosine monophosphate-activated protein kinase α2 (AMPKα2) (*N* = 5). (**c**) Phosphorylated acetyl coenzyme A carboxylase (pACC) (*N* = 5). (**d**) Peroxisome proliferator-activated receptor gamma coactivator-1α (PGC-1α) (*N* = 5). (**e**) Carnitine palmitoyltransferase-1β (CPT-1β) (*N* = 5). (**f**) Cluster of differentiation 36 (CD36) (*N* = 5). Densitometry was normalized to glyceraldehyde 3-phosphate dehydrogenase (GAPDH) as an internal control.

**Figure 3 ijms-22-01177-f003:**
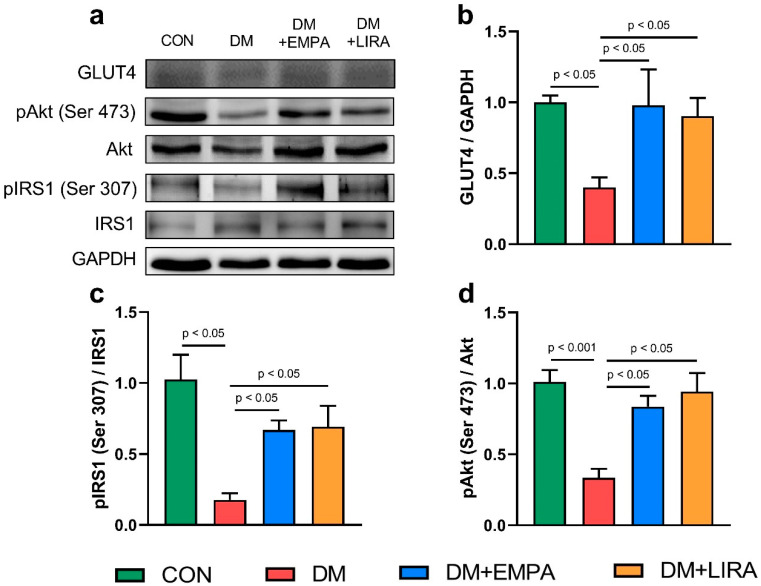
Expression of cardiac glucose metabolism regulatory proteins in control (CON) and diabetic (DM) rats with and without treatment with empagliflozin (EMPA) and liraglutide (LIRA). (**a**) Representative immunoblot image. (**b**) Average data of glucose transporter 4 (GLUT4) (*N* = 5). (**c**) Ratio of phosphorylated insulin receptor substrate 1 (pIRS1) (Ser 307) to insulin receptor substrate 1 (IRS1) (*N* = 5). (**d**) Ratio of phosphorylated protein kinase B (pAkt) (Ser 473) to protein kinase B (Akt) (*N* = 5). Densitometry was normalized to glyceraldehyde 3-phosphate dehydrogenase (GAPDH) as an internal control.

**Figure 4 ijms-22-01177-f004:**
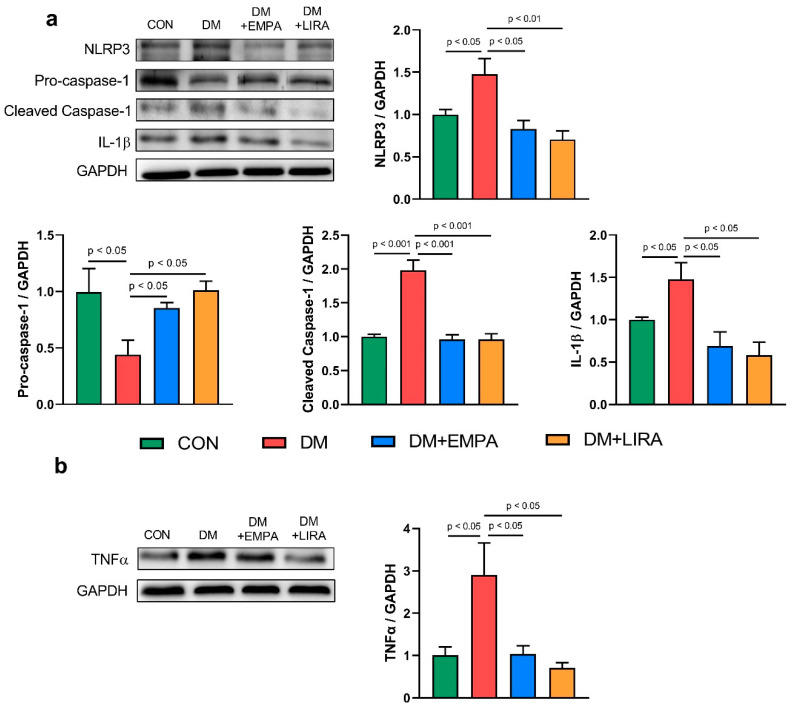
Expression of cardiac inflammatory proteins in control (CON) and diabetic (DM) rats with and without treatment with empagliflozin (EMPA) and liraglutide (LIRA). (**a**) Representative immunoblot image and average data of NOD-like receptor family pyrin domain containing 3 (NLRP3) (*N* = 4). Pro-caspase-1 (*N* = 4). Cleaved caspase-1 (*N* = 4) and interleukin (IL)-1β (*N* = 4). (**b**) Representative immunoblot image and average data of tumor necrosis factor-α (TNF-α) (*N* = 5). Densitometry was normalized to glyceraldehyde 3-phosphate dehydrogenase (GAPDH) as an internal control.

**Figure 5 ijms-22-01177-f005:**
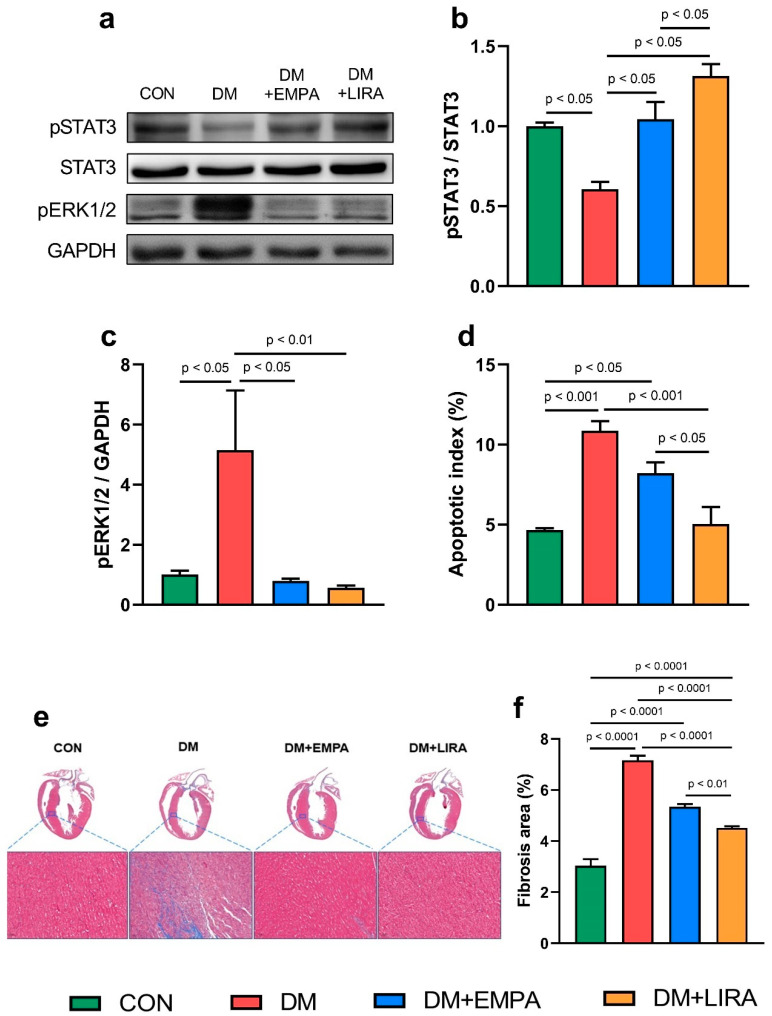
Cardiac fibrosis and apoptosis in control (CON) and diabetic (DM) rats with and without treatment with empagliflozin (EMPA) and liraglutide (LIRA). (**a**) Representative immunoblot image. (**b**) Ratio of phosphorylated signal transducer and activator of transcription 3 (pSTAT3) to signal transducer and activator of transcription 3 (STAT3) (*N* = 5). (**c**) Phosphorylated extracellular signal-regulated kinase 1/2 (pERK1/2) (*N* = 5). Densitometry was normalized to glyceraldehyde 3-phosphate dehydrogenase (GAPDH) as an internal control (*N* = 5). (**d**) Apoptotic index (*N* = 5). (**e**) Histological section and Masson’s trichrome staining and (f) percentage of areas exhibiting fibrosis (*N* = 5).

**Figure 6 ijms-22-01177-f006:**
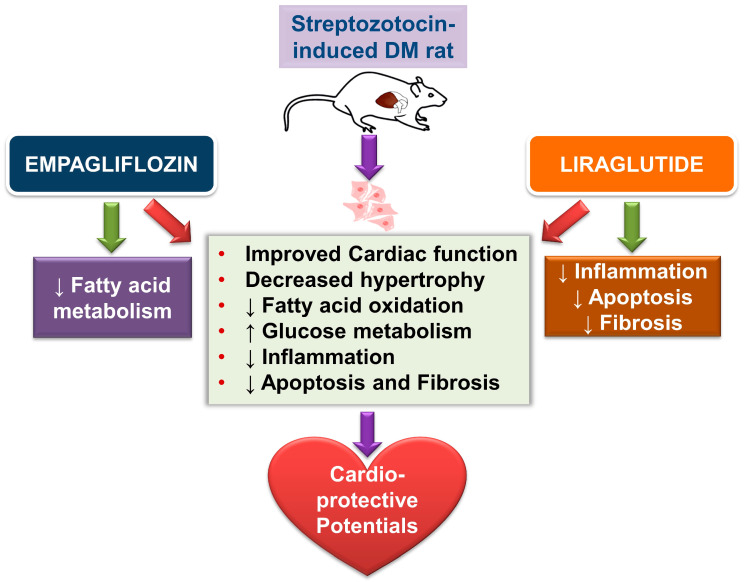
Schematic illustration of the proposed effects of empagliflozin and liraglutide in diabetic (DM) hearts. Empagliflozin and liraglutide may improve the remodeling of DM hearts through multiple effects: decreased cardiac hypertrophy; switching of fatty acid and glucose metabolism; and anti-inflammatory, anti-fibrosis, and apoptosis effects.

**Table 1 ijms-22-01177-t001:** Physical characteristics of control (CON), diabetes mellitus (DM), empagliflozin-treated DM (DM + EMPA), and liraglutide-treated DM (DM + LIRA) rats.

	CON	DM	DM + EMPA	DM+LIRA
**FBS (mg/dL)**				
2 weeks after STZ injection	109.8 ± 1.1	398.6 ± 11.7 *	394.5 ± 8.9 *	395.9 ± 3.6 *
4 weeks after treatment	113 ± 2.9	429.5 ± 20.2 *	151.8 ± 10.4 ^#^	284.9 ± 25.1 *^#^^§^
**Biochemical values after treatment**				
TG (mg/dL)	84.9 ± 6.0	144.3 ± 25.5	74.4 ± 19.9 ^#^	98.9 ± 13.9
TC (mg/dL)	52.8 ± 3.6	60.6 ± 4.2	50.5 ± 3.7	52.3 ± 2.9
HDL-C (mg/dL)	22.8 ± 1.1	27.9 ± 1.7 *	23.5 ± 0.8	26.2 ± 1.5
LDL-C (mg/dL)	4.1 ± 0.4	6.3 ± 0.9 *	3.9 ± 0.2 ^#^	2.8 ± 0.3 ^#^
FFAs (µg/dL)	0.7 ± 0.0	1.0 ± 0.2 *	0.6 ± 0.0 ^#^	0.6 ± 0.1 ^#^
ST2 (pg/mL)	296.2 ± 39.1	455.1 ± 54.7 *	292.5 ± 18.7 ^#^	295.3 ± 42.9 ^#^
**BP and HR after treatment**				
SBP (mmHg)	127.8 ± 5.5	113.5 ± 5.4	118.3 ± 11.7	116.5 ± 6.9
DBP (mmHg)	73.9 ± 4.4	65.3 ± 5.3	75.9 ± 9.3	60.5 ± 6.4
HR (bpm)	440 ± 13.9	311.6 ± 19.7 *	367.1 ± 24.6	387.4 ± 21.0
**Weight**				
BW 2 weeks after STZ injection (g)	380.4 ± 4.5	368.0 ± 14.2	368.8 ± 9.5	369.0 ± 6.6
BW after treatment (g)	543.8 ± 5.6	379.1 ± 10.2 *	441.9 ± 17.3 *^#^	402.9 ± 15.5 *
HW (g)	1.6 ± 0.0	1.4 ± 0.0 *	1.6 ± 0.0 ^#^	1.3 ± 0.0 *^§^
HW/BW (mg/g)	3.2 ± 0.0	4.2 ± 0.2 *	3.6 ± 0.1 ^#^	3.3 ± 0.1 ^#^

FBS, fasting blood glucose; STZ, streptozotocin; TG, triglyceride; TC, total cholesterol; HDL-C, high-density lipoprotein cholesterol; LDL-C, low-density lipoprotein cholesterol; FFAs, free fatty acids; ST2, suppression of tumorgenicity-2; BP, blood pressure; SBP, systolic blood pressure; DBP, diastolic blood pressure; HR, heart rate; bpm, beats per minute; BW, body weight; HW, heart weight. Number of rats *N* = 8 per group. Values are expressed as the mean ± standard error of the mean (SEM). ** p* < 0.05 vs. the control; ^#^
*p* < 0.05 vs. DM; ^§^
*p* < 0.05 vs. DM + EMPA.

**Table 2 ijms-22-01177-t002:** Echocardiograms, physical characteristics, biochemical values, and histopathology of empagliflozin-treated diabetes mellitus (DM) rats (DM + EMPA), liraglutide-treated DM rats (DM + LIRA), and diabetes treated with empagliflozin and liraglutide rats (DM + EMPA + LIRA).

	DM + EMPA	DM + LIRA	DM + EMPA + LIRA
**Echocardiograms**			
LVEDd (mm)	6.9 ± 0.5	7.1 ± 0.3	6.7 ± 0.0
LVESd (mm)	3.4 ± 0.5	3.7 ± 0.1	3.2 ± 0.2
EDV (mL)	0.8 ± 0.1	0.9 ± 0.1	0.7 ± 0.1
ESV (mL)	0.09 ± 0.0	0.14 ± 0.0 ^§^	0.07 ± 0.0 ^¶^
IVSd (mm)	1.9 ± 0.2	1.7 ± 0.1	1.9 ± 0.0
LVPW (mm)	2.2 ± 0.1	1.8 ± 0.1 ^§^	2.2 ± 0.1 ^¶^
EF (%)	85.4 ± 3.9	81.2 ± 2.2	89.3 ± 1.9
FS (%)	48.8 ± 4.4	45.1 ± 2.4	57.4 ± 2.1 ^¶^
**Physical characteristics**			
HR (bpm)	407.6 ± 22.6	413.4 ± 5.8	424 ± 8.6
SBP (mmHg)	119.8 ± 7.8	115.8 ± 6.8	118.8 ± 4.5
DBP (mmHg)	71.0 ± 6.4	61.4 ± 9.2	68.8 ± 3.2
BW 2 weeks after STZ injection (g)	355.4 ± 9.6	358.4 ± 4.9	356.8 ± 3.2
BW after treatment (g)	416.8 ± 17.1	327.8 ± 8.4 ^§^	399 ± 12.6^¶^
HW (g)	1.6 ± 0.0	1.5 ± 0.1	1.6 ± 0.1
HW/BW (mg/g)	3.9 ± 0.3	4.4 ± 0.2	3.6 ± 0.2
**Biochemical values**			
FBS 2 weeks after STZ injection (mg/dL)	396.0 ± 12.4	396.6 ± 5.7	397.2 ± 12.8
FBS after treatment (mg/dL)	142.4 ± 21.5	331.6 ± 22.4 ^§^	109.6 ± 14.5 ^¶^
TG (mg/dL)	60.2 ± 16.8	86.8 ± 15.6	47.0 ± 4.8
TC (mg/dL)	51.8 ± 3.4	55.0 ± 3.4	55.5 ± 4.6
HDL-C (mg/dL)	23.2 ± 1.3	26.0 ± 1.0	23.6 ± 1.0
LDL-C (mg/dL)	3.8 ± 0.4	3.0 ± 0.3	3.8 ± 0.2
FFAs (µg/dL)	0.6 ± 0.0	0.7 ± 0.0	0.6 ± 0.0
ST2 (pg/mL)	290.2 ± 17.8	296.2 ± 38.2	281.0 ± 21.2
**Histopathology**			
Fibrosis area (%)	5.3 ± 0.1	4.5 ± 0.1 ^§^	4.3 ± 0.1 ^§^

LVEDd, left ventricular end-diastolic diameter; LVESd, left ventricular end-systolic diameter; EDV, end-diastolic volume; ESV, end-systolic volume; IVSd, interventricular septum end diastole; LVPW, left ventricular posterior wall; EF, ejection fraction; FS, fractional shortening; HR, heart rate; bpm, beats per minute; SBP, systolic blood pressure; DBP, diastolic blood pressure; BW, body weight; HW, heart weight; STZ, streptozotocin; FBS, fasting blood glucose; TG, triglyceride; TC, total cholesterol; HDL-C, high-density lipoprotein cholesterol; LDL-C, low-density lipoprotein cholesterol; FFAs, free fatty acids; ST2, suppression of tumorgenicity-2. Number of rats *N* = 5 per group. Values are expressed as the mean ± SEM. ^§^
*p* < 0.05 vs. DM + EMPA; ^¶^
*p* < 0.05 vs. DM + LIRA.

## Data Availability

The data presented in this study are available in the article.

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
