# Peer review of "Empagliflozin and Liraglutide Differentially Modulate Cardiac Metabolism in Diabetic Cardiomyopathy in Rats"

_ijms, 2021, doi:10.3390/ijms22031177_

Round 1

Reviewer 1 Report

In the present rat model, the Authors investigated the impacts of GLP1RAs and or SGLT2is on myocardial energy metabolism, cardiac function, and apoptosis signaling in male Wistar rats with streptozotocin induced diabetes. After 4 weeks of treatment, empagliflozin and liraglutide normalized myocardial dysfunction evaluated by echocardiograms in diabetic rats and attenuated myocardial fibrosis and apoptosis. Empagliflozin modulated fatty acid and glucose metabolism, while liraglutide regulated inflammation and apoptosis in diabetic cardiomyopathy.

The paper is well written, the aim is an interesting one and there are note of novelty in the results. The methodological approach is appropriate, the results are adequately commented upon

However, the following concerns should be addressed

1) Limitations should be reviewed and commented on in the Discussion section.

2) For reasons of readability, the Results and Discussion sections should be shortened.

Author Response

Dear Reviewer 1,

Thank you very much for your detailed comments. The comments are very instructive and very helpful to this manuscript. The responses to the comments are enumerated below:

Point 1: Regarding the comment (1): “ Limitations should be reviewed and commented on in the Discussion section”.

Response 1: Thank you very much for your comment. According to your suggestions, we have included the study limitations in the revised discussion section as follows: “This study has some limitations. First, our results showed that the combined treatment with GLP-1RA and SGLT2i may lead to a potential novel strategy in targeting DCM, but the effects of empagliflozin and liraglutide on the myocardial ATP content, cytokine analysis, NLRP3 signaling, cardiac metabolism, and collagen expressions in DCM remain unclear. Besides, the mechanism underlying the superiority of the combined treatment needs further study. Second, our findings may not fully translate to DM patients in clinics since the treatment already started 2 weeks after STZ injection. Third, the animal numbers used for this study were relatively fewer and the power of statistics may be underestimated, even though Duncan’s method was chosen for post hoc analysis. Finally, we investigated the local inflammatory responses by using Western blot analysis, but not ELISA to measure the cardiac expression of inflammatory cytokine in DCM. Thus, our results may be relatively semi-quantitative.” (page 12, lines 268-278; red font)

Point 2: Regarding the comment (2): For reasons of readability, the Results and Discussion sections should be shortened.

Response 2: We appreciate this comment very much. According to your suggestions, we have shortened the Results and Discussion in our revised manuscript.

The above descriptions are the responses to your comments and suggestions. We highly appreciated your careful and detailed review of our manuscript. Thank you very much.

Sincerely yours,

Ting-I Lee, MD, PhD

Director, Department of Internal Medicine, Division of Endocrinology and Metabolism, School of Medicine, College of Medicine, Taipei Medical University.

Reviewer 2 Report

The Authors have investigated the effect of two antidiabetic drugs, empagliflozin andliraglutide, in a rat model of cardiomyopathy. Please find below my comments.

METHODS:

In the animal model section, the authors affirm that the animals were randomly assigned to receive empagliflozin or liraglutide or a combined treatment with empagliflozin and liraglutide. However:

- in the Table 1 the basal values are already reported for 4 groups, but at the baseline the groups did not exist yet! It would be important to know the animal data after 2 weeks from STZ injection, before randomization. Could be useful to put these data in the table instead of the basal values.

-in the TABLE 1/figures there are not data about the group of the combined treatment. It would be interesting, in my opinion, to also show these results.

In addition, the authors should better explain how empagliflozin was administered to the animals.

The authors should explain the choice of the statistical test, the  Duncan’s method.

RESULTS:

The authors focus their attention on insulin resistance. In my opinion in a model of Type 1 diabetes this is not a crucial point. I think that the authors have to better explain this point and improve the analysis. For example:

-  the phosphorylated sites in the insulin signalling targets evaluated are not indicated. The Authors should know the crucial effect of different sites of phosphorylation.

-the Authors show an increased expression of GLUT4. However, an increased expression is not enough to hypothesize an increased GLUT4 expression on the cell membrane, the membrane translocation should be analysed.

In addition, phosphorylated targets should be investigated in relation to their total protein (notphosphorylated protein) and this evaluation has to be perform on the same membrane. The Authors investigated the ph- and unphosphorylated target just forSTAT3 and AMPK but they show result achieved on different membrane and not inthe same one. Th data should be expressed as ph/total target, otherwise is difficult to understand the results, for example: the phosphorylated AMPK protein and the total one have the same modulation..so, are there not differences among the groups on ph AMPKs/AMPK ratio?. ..Also theNLRP3 inflammasome pathway should be evaluated on the same samples and on thesame membrane.

Furthermore, in my opinion the cytokines should be evaluated with a quantitative assay, i.e. ELISA assay.

DISCUSSION:

I do not agree with the following sentence: "..but empagliflozin even normalized the blood glucose level  in DM rats": in the table the level of FBS of the empagliflozin group is still statistically different from the control. In the discussion the authors affirm:” These findings suggest that amelioration of insulin sensitivity by liraglutide and empagliflozin may virtually be caused by their added effect on insulin signaling transduction.” But I do not understand how the authors demonstrated that the two drugs improved insulin sensitivity,there is not a OGTT assay in the paper, the authors just show that the drugs affect the insulin signalling pathway...In my opinion it is not proper to refer to these antidiabetic drugs as an anti-inflammatory agents.

I do not agree with the title of the Figure 6 and the relative sentence reported in the discussion:"... mechanism of action of empagliflozin and  liraglutide in diabetic (DM) hearts.": these are not mechanisms of action, but effects associated with drug administration.

Finally, as the treatment started already after 2 weeks from STZ injection, could these drugs be useful to prevent cardiomyopathy rather than to its treatment?

Author Response

Dear Reviewer 2,

Thank you very much for your detailed comments. The comments are very instructive and very helpful to this manuscript. The responses to the comments are enumerated below:

Point 1: Regarding the comments: “In the animal model section, the authors affirm that the animals were randomly assigned to receive empagliflozin or liraglutide or a combined treatment with empagliflozin and liraglutide. However: - in the Table 1 the basal values are already reported for 4 groups, but at the baseline the groups did not exist yet! It would be important to know the animal data after 2 weeks from STZ injection, before randomization. Could be useful to put these data in the table instead of the basal values.

Response 1: We appreciate this comment very much. According to your suggestions, we have added the values of FBS and BWs 2 weeks after STZ injection and removed the basal values in Table1. (Page 2-3, red font).

Point 2: Regarding the comment: “ -in the TABLE 1/figures there are not data about the group of the combined treatment. It would be interesting, in my opinion, to also show these results.

Response 2: Thank you very much for your comment. According to your suggestion, we have provided the data of the combined treatment group in Table 2 (page 10, red font) and the revised results as follows: “There was no difference in the heart rates, BP, HWs, heart-to-body weight ratios and the biochemical values of the combined-treated DM group with the groups treated with the individual compounds. The combined-treated DM rats had higher BWs and lower FBS than liraglutide-treated DM rats.”  (page 10, lines 194-197, red font). However, some data about the combined treatment remains unclear in this study, we have discussed this in our limitation (page 12, lines 268-272; red font) in the revised discussion as follows: “Our results showed that the combined treatment with GLP-1RA and SGLT2i may lead to a potential novel strategy in targeting DCM, but the effects of empagliflozin and liraglutide on the myocardial ATP content, cytokine analysis, NLRP3 signaling, cardiac metabolism, and collagen expressions in DCM remain unclear. Besides, the mechanism underlying the superiority of the combined treatment needs further study.”

Point 3: Regarding your comments “In addition, the authors should better explain how empagliflozin was administered to the animals.”

Response 3: Thank you very much for this comment. We are very sorry for the unclear presentation of animal preparation. We have corrected it in the revised manuscript “…empagliflozin (10 mg/kg/day, oral gavage),” (page 1, line 31, red font) and, “... DM rats were randomly assigned to receive empagliflozin (10 mg/kg/day, oral gavage…” (page 13 line 299, red font).

Point 4: Regarding the comments: “The authors should explain the choice of the statistical test, the Duncan’s method.”

Response 4: Thank you so much for your comment. Duncan's multiple range test provides significance levels for the difference between any pair of means, regardless of whether a significant F resulted from an initial analysis of variance. (Bewick, V et al. Critical care (London, England) 2004; 8, 130-136). As we used the one-way ANOVA test to compare the mean difference between groups with a small amount number of the animal so Duncan’s method was chosen to avoid underestimate the power of statistics as our previous papers (Lee TI et al, Int. J. Mol. Sci. 2019, 20(7), 1680). We have explained that in the revised limitation (page 12, lines 274-275; red font) as follows: “The animal numbers used for this study was relatively fewer and the power of statistics may be underestimated, even though Duncan’s method was chosen for post hoc analysis.”

Point 5: Regarding the comments: “The authors focus their attention on insulin resistance. In my opinion in a model of Type 1 diabetes this is not a crucial point. I think that the authors have to better explain this point and improve the analysis. For example:-  the phosphorylated sites in the insulin signalling targets evaluated are not indicated. The Authors should know the crucial effect of different sites of phosphorylation.”

Response 5: We appreciate this comment very much. We agree with your comment that insulin resistance is not a crucial point in a model of Type 1 diabetes and the phosphorylated sites in the insulin signaling targets evaluated are not indicated. According to your suggestion, we have explained this point in the revised discussion. Moreover, we used the phosphorylation of Ser307 on rat IRS-1 which inhibits the tyrosine phosphorylation of IRS-1 by the insulin receptor and impairs metabolic insulin signaling pathways (Gao Z et al. J Biol Chem. 2002; 277: 48115-21). We have clarified it in the revised Figure 3 (page 6, lines 145,148 and 154-155),  revised Methods (pages 14, line 350-351, red font) and revised discussion (pages 11, lines 240-243, red font) as follows: “Although insulin resistance is not a crucial point in a model of Type 1 diabetes, this study found that DM hearts treated with liraglutide or empagliflozin exhibited restored pAkt (Ser 473) and pIRS1 (Ser 307) protein levels, which may impair metabolic insulin signaling pathways. These effects might contribute to the improvement of cardiac function and metabolic homeostasis [25].”

Point 6: Regarding the comments: “-the Authors show an increased expression of GLUT4. However, an increased expression is not enough to hypothesize an increased GLUT4 expression on the cell membrane, the membrane translocation should be analysed.”

Response 6: Thank you very much for this comment. We agreed with your comment that an increased expression is not enough to hypothesize an increased GLUT4 expression on the cell membrane. According to your suggestion, we have discussed that in the revised manuscript (page 11, lines 243-247, red font) as follows: “Our results showed that the GLUT4 protein expression significantly increased with empagliflozin or liraglutide treatment compared to the DM rat hearts. However, an increased protein level is not enough to hypothesize an increased GLUT4 expression on the cell membrane, since the membrane translocation was not analyzed in this study.”

Point 7: Regarding the comments: “In addition, phosphorylated targets should be investigated in relation to their total protein (notphosphorylated protein) and this evaluation has to be perform on the same membrane. The Authors investigated the ph- and unphosphorylated target just for STAT3 and AMPK but they show result achieved on different membrane and not inthe same one. Th data should be expressed as ph/total target, otherwise is difficult to understand the results, for example: the phosphorylated AMPK protein and the total one have the same modulation..so, are there not differences among the groups on ph AMPKs/AMPK ratio? Also the NLRP3 inflammasome pathway should be evaluated on the same samples and on the same membrane.”

Response 7: We appreciate this comment very much. According to your suggestions, we evaluated pAMPKa2/AMPKa2, NLRP3 signaling and pSTAT3/STAT3 on the same membrane and have changed the presentation of our data in our revised results and figures as follows: Page 4, lines 123-125, red font “Both empagliflozin-treated DM rats and liraglutide-treated DM rats had higher ratio protein expressions of phosphorylated 5' adenosine monophosphate-activated protein kinase α2 (pAMPKa2) to AMPKa2 compared to DM rats (Figure 2b).” and page 5, lines 133, 136-137, red font “Figure 2… (b) Ratio of phosphorylated adenosine monophosphate-activated protein kinase α2 (pAMPKa2) to AMPKa2 (N=5).”.  Figure 4 (page 7, lines 164-170). Page 8 lines 172-175, red font “The ratio of phosphorylated signal transducer and activator of transcription 3 (pSTAT3) to STAT3 (Figure 5b) were significantly higher in empagliflozin-treated and liraglutide-treated DM groups than the DM group. Moreover, liraglutide-treated DM rats had a higher pSTAT3/STAT3 compared to empagliflozin-treated DM rats.” and “Figure 5… (b) Ratio of phosphorylated signal transducer and activator of transcription 3 (pSTAT3) to STAT3 (N=5).” (page 9, lines 183, 186-187, red font).

Point 8: Regarding the comments: “Furthermore, in my opinion the cytokines should be evaluated with a quantitative assay, i.e. ELISA assay.”

Response 8: We appreciate this comment very much. We agree with your comment that cytokines should be evaluated with a quantitative assay, i.e. ELISA assay. According to your suggestions, we have discussed this in our limitation in the revised discussions (page 12, lines 276-278, red font) as follows: “We investigated the local inflammatory responses by using Western blot analysis, but not ELISA to measure the cardiac expression of inflammatory cytokine DCM. Thus, our results may be relatively semi-quantitative.”

Point 9: Regarding the comments: “- I do not agree with the following sentence: "..but empagliflozin even normalized the blood glucose level in DM rats": in the table the level of FBS of the empagliflozin group is still statistically different from the control.”

Response 9: We agree with your comment. According to your suggestion, we have revised the sentence as follow: Our study compared the action effects of empagliflozin and liraglutide which belong to two new classes of anti-hyperglycemic agents, as cardio-protection in DCM, and we found that both empagliflozin and liraglutide could effectively control blood glucose.” in the revised manuscript (page 10, line 213; and page 11, lines 214-216, red font).

Point 10: Regarding the comments: “-- In the discussion the authors affirm:” These findings suggest that amelioration of insulin sensitivity by liraglutide and empagliflozin may virtually be caused by their added effect on insulin signaling transduction.” But I do not understand how the authors demonstrated that the two drugs improved insulin sensitivity, there is not a OGTT assay in the paper, the authors just show that the drugs affect the insulin signalling pathway...In my opinion it is not proper to refer to these antidiabetic drugs as an anti-inflammatory agents.”

Response 10: Thank you very much for your comment. We agree with your comments and deleted these sentences in the revised discussion.

Point 11: Regarding the comments: “I do not agree with the title of the Figure 6 and the relative sentence reported in the discussion:"... mechanism of action of empagliflozin and liraglutide in diabetic (DM) hearts.": these are not mechanisms of action, but effects associated with drug administration.

Response 11: Thank you very much for your comment. According to your suggestion, we have changed the title of Figure 6 to: “Schematic illustration of the proposed action effects of empagliflozin and liraglutide in diabetic (DM) hearts” (page 12, lines 285-286, red font) and revised the sentence in the discussion (page 11, lines 264-265 and page 12, line 266, red font) as follow: “Figure 6 demonstrates the potential action effects of empagliflozin (SGLT2i) and liraglutide (GLP-1RA) in DM hearts in this study.”

Point 12: Regarding the comments: “Finally, as the treatment started already after 2 weeks from STZ injection, could these drugs be useful to prevent cardiomyopathy rather than to its treatment?

Response 12: Thank you very much for your comment. We have discussed this limitation in the revised discussion (page 12, lines 272-274, red font) as follows:  “Our findings may not fully translate to DM patients in clinics since the treatment already started 2 weeks after STZ injection.”

The above descriptions are the responses to your comments and suggestions. We highly appreciated your careful and detailed review on our manuscript. Thank you very much.

Sincerely yours,

Ting-I Lee, MD, PhD

Director, Department of Internal Medicine, Division of Endocrinology and Metabolism, School of Medicine, College of Medicine, Taipei Medical University.

Round 2

Reviewer 1 Report

The authors have dealt adequately with mine concerns.

Author Response

Response to Reviewer 1

Dear Reviewer 1,

We highly appreciated your careful and detailed review of our manuscript.

Thank you very much.

Sincerely yours,

Ting-I Lee, MD, PhD

Director, Department of Internal Medicine, Division of Endocrinology and Metabolism, School of Medicine, College of Medicine, Taipei Medical University.

Reviewer 2 Report

I appreciate the changes done by the authors. However, I still have some criticisms.

-Table 1. I appreciated the changes done by the authors. However, I do not understand the statistical analysis on FBS -4 weeks after treatment. The stars means that the value is different from the control, so the stars should be present in both empagliflozin and liraglutide groups, or just in the liraglutide group if the empagliflozin level of FBS is not different from the control (in this case the star must be removed). It does not make sense the absence of the star in the liraglutide group.

I appreciated the modified sentence present in this version of the article (page 10, line 213; and page 11, lines 214-216). I think could be improved by underlying the difference of the effect on the base of the table 1: the FBS value of the liraglutide group is statistically different from the FBS value of the empagliflozin group, so empagliflozin seems to be more efficient to reduce glycemia.

-I appreciated the changes performed by the authors about insulin resistance. However, the phosphorylation of IRS-1 in Ser307 impairs the insulin signalling, as correctly reported by the authors in the cover letter. However, the results show an increased level of phosphorylated IRS-1 in Ser 307 in the control group and in animals treated with the drugs in comparison to DM...

In addition, the sentence “which may impair metabolic insulin signalling pathways”  (discussion, line 241-242) seems to be related to the restored pAkt (Ser 473) and pIRS1 (Ser 307) protein levels and this is not correct.

Furthermore, the best way to report the results in my opinion is to show the expression of phosphorylated/unphosphorylated target more that ph-target/GADPH.

-I am sorry but I do not well understand the results about AMPK. I would expected that the activated form of AMPK induced a phosphorylation of ACC…but the results reported in this paper show an opposite modulation...

In addition, the authors have to better explain the meaning of the phosphorylation of STAT in their contest, because an increase of the phosphorylated form of STAT is not always a positive effect on diabetes.

-In the figure 4, the authors presented the data about the NLRP3 inflammasome signaling. The graph related to the ratio of cleaved caspase/GAPDH shows a different and inconsistent result in comparison to the related WB band. In fact, in the WB images the band of the cleaved caspase is similar between control and diabetes...(and it means that NLRP3 inflammasome is not differently activated between control and diabetic animals). Moreover, TNF is not linked to NLRP3 signaling, so it does not make sense showing the data all together.

-In contrast with the text, in the table 2 the statistical analysis shows that the fibrosis area of the combined group is statistically different in comparison to the liraglutide group.

-Regarding Figure 6 and the relative sentence in the discussion, I appreciated the changes done by the authors, if it possible I would delete “action” form the sentences..i.e. "Figure 6 demonstrates the potential effects of empagliflozin (SGLT2i) and liraglutide (GLP-1RA) in DM hearts in this study.”

Author Response

Response to Reviewer 2

Dear Reviewer 2,

Thank you very much for your detailed comments. The comments are very instructive and very helpful to this manuscript. The responses to the comments are enumerated below:

Point 1: Regarding the comment: “Table 1. I appreciated the changes done by the authors. However, I do not understand the statistical analysis on FBS -4 weeks after treatment. The stars means that the value is different from the control, so the stars should be present in both empagliflozin and liraglutide groups, or just in the liraglutide group if the empagliflozin level of FBS is not different from the control (in this case the star must be removed). It does not make sense the absence of the star in the liraglutide group.”

Response 1: We appreciate this comment very much. We are very sorry for the wrong position of the star which represents the difference from the control group in Table 1. According to your suggestions, we have added the star in the liraglutide group and removed it in the empagliflozin group in the revised Table 1. (Page 2, red font).

Point 2: Regarding the comment: “I appreciated the modified sentence present in this version of the article (page 10, line 213; and page 11, lines 214-216). I think could be improved by underlying the difference of the effect on the base of the table1: the FBS value of the liraglutide group is statistically different from the FBS value of the empagliflozin group, so empagliflozin seems to be more efficient to reduce glycemia.”

Response 2: We appreciate this comment very much. According to your suggestion, we have modified the sentence in the revised manuscript (page 9, line 215 and page 10, lines 216-219, red font) as follows: “Our study compared the effects of empagliflozin and liraglutide which belong to two new classes of anti-hyperglycemic agents, as cardio-protection in DCM, and we found that the FBS value of the liraglutide group is statistically different from the FBS value of the empagliflozin group, so empagliflozin seems to be more efficient to reduce glycemia.”

Point 3: Regarding your comments: “I appreciated the changes performed by the authors about insulin resistance. However, the phosphorylation of IRS-1 in Ser307 impairs the insulin signalling, as correctly reported by the authors in the cover letter. However, the results show an increased level of phosphorylated IRS-1 in Ser 307 in the control group and in animals treated with the drugs in comparison to DM... In addition, the sentence “which may impair metabolic insulin signalling pathways” (discussion, line 241-242) seems to be related to the restored pAkt (Ser 473) and pIRS1 (Ser 307) protein levels and this is not correct. Furthermore, the best way to report the results in my opinion is to show the expression of phosphorylated/unphosphorylated target more that ph-target/GADPH.”

Response 3: We appreciate this comment very much. According to your suggestion, we deleted the sentence “which may impair metabolic insulin signalling pathways” and revised our sentence (page 11, lines 246-248, red font) as follow: “This study found that DM hearts treated with liraglutide or empagliflozin exhibited restored the ratios of pIRS1 (Ser 307)/IRS1 and pAkt (Ser 473)/Akt. However, it is not clear whether these effects may play a role in cardiac function and metabolic homeostasis in these animals

Furthermore, according to your suggestion, we have changed the presentation of our data in the revised Figure 3 (page 6), and the results as follows:  Page 6, lines 145-148, red font “the ratio of phosphorylated insulin receptor substrate 1 (pIRS1) (Ser 307) to IRS1 was significantly higher in empagliflozin-treated DM rats and liraglutide-treated DM rats than in DM rats (Figure 3c). Similarly, both empagliflozin-treated DM rats and liraglutide-treated DM rats had increased the ratio of phosphorylated protein kinase B (pAkt) (Ser 473) to Akt compared to DM rats” and page 6, lines 153-155, red font: “ (c) Ratio of phosphorylated insulin receptor substrate 1 (pIRS1) (Ser 307) to IRS1 (N=5). (d) Ratio of phosphorylated protein kinase B (pAkt) (Ser 473) to Akt (N=5).”

Point 4: Regarding your comments: “I do not well understand the results about AMPK. I would expected that the activated form of AMPK induced a phosphorylation of ACC…but the results reported in this paper show an opposite modulation...In addition, the authors have to better explain the meaning of the phosphorylation of STAT in their contest, because an increase of the phosphorylated form of STAT is not always a positive effect on diabetes.”

Response 4: Thank you very much for this comment. We agree with your comment and we added the sentence in our revised manuscript (page 11, lines 232-236, red font) as follows: “The activated form of AMPK was expected to induce phosphorylation of ACC. However, our results showed an opposite modulation. The previous study also has shown an activated AMPK with suppressed ACC activity in metformin-treated hepatocytes (Zhou G et al. J Clin Invest. 2001;108:1167-1174). Additional studies will be required to further elucidate the precise mechanism underlying for our findings.”

We also agree with your comment that the authors have to better explain the meaning of the phosphorylation of STAT. We added the following sentences in our revised manuscript as follows “Activation of STAT3 has been suggested to reduce cardiac fibrosis or hypertrophy in DM via the inhibition of apoptosis or increase of anti-oxidants (Pipicz M. et al. Int J Mol Sci. 2018;19:3572), although STAT3 may promote proliferation and collagen production in isolated cardiac fibroblasts treated with a high concentration of glucose (Harhous Z et al. Front Cardiovasc Med 2019, 6, 150). In this study, consistent with that in the previous studies (Wang C et al. Cell Physiol. Biochem. 2018;45:2107–2121, Xue R et al.Clin Sci (Lond). 2016 Mar; 130(5):377-92.), DM rats had decreased p-STAT3 as compared to control. Moreover, an increased in the ratio of pSTAT3/STAT3 protein expression in empagliflozin-treated DM rats and liraglutide-treated DM rats. These findings suggest the potential role of STAT3 regulation in cardiac remodeling in empagliflozin or liraglutide-treated DM rats.” (page 10, lines 264-267 and page 11, lines 268-271).

Point 5: Regarding the comments: “In the figure 4, the authors presented the data about the NLRP3 inflammasome signaling. The graph related to the ratio of cleaved caspase/GAPDH shows a different and inconsistent result in comparison to the related WB band. In fact, in the WB images the band of the cleaved caspase is similar between control and diabetes...(and it means that NLRP3 inflammasome is not differently activated between control and diabetic animals). Moreover, TNF is not linked to NLRP3 signaling, so it does not make sense showing the data all together.”

Response 5: Thank you so much for your comment. According to your suggestion, we have presented typical WB band of Cleaved Caspase-1 in the revised Figure 4a and moved the data of TNF-α to the revised Figure 4b.

Point 6: Regarding the comments: “In contrast with the text, in the table 2 the statistical analysis shows that the fibrosis area of the combined group is statistically different in comparison to the liraglutide group.”

Response 6: We appreciate this comment very much. We are very sorry for the wrong marker presents the statistical difference of fibrosis area of the combined group with the empagliflozin group (not liraglutide group) in Table 2. According to your suggestions, we have modified the marker § in the combined group in revised Table 2 (Page 9, red font) which had a similar meaning with the sentence: “Moreover, the fibrosis area of combine-treated DM rats was similar to that in liraglutide-treated DM rats but lesser than that of empagliflozin-treated DM rats.” (page 9, lines 199-201).

Point 7: Regarding the comments: “Regarding Figure 6 and the relative sentence in the discussion, I appreciated the changes done by the authors, if it possible I would delete “action” form the sentences..i.e. "Figure 6 demonstrates the potential effects of empagliflozin (SGLT2i) and liraglutide (GLP-1RA) in DM hearts in this study.””

Response 7: Thank you very much for your comment. According to your suggestion, we have changed the title of Figure 6 to: “Schematic illustration of the proposed effects of empagliflozin and liraglutide in diabetic (DM) hearts” (page 12, lines 297-298, red font) and revised the sentence in the discussion (page 11, line 276-278, red font) as follow: “Figure 6 demonstrates the potential effects of empagliflozin (SGLT2i) and liraglutide (GLP-1RA) in DM hearts in this study.”

The above descriptions are the responses to your comments and suggestions. We highly appreciated your careful and detailed review of our manuscript. Thank you very much.

Sincerely yours,

Ting-I Lee, MD, PhD

Director, Department of Internal Medicine, Division of Endocrinology and Metabolism, School of Medicine, College of Medicine, Taipei Medical University.